# Towards a Unified Framework of Contrastive Learning for Disentangled Representations

**Stefan Matthes, Zhiwei Han, Hao Shen**

fortiss GmbH, Munich, Germany

`{matthes,han,shen}@fortiss.org`

## Abstract

Contrastive learning has recently emerged as a promising approach for learning data representations that discover and disentangle the explanatory factors of the data. Previous analyses of such approaches have largely focused on individual contrastive losses, such as noise-contrastive estimation (NCE) and InfoNCE, and rely on specific assumptions about the data generating process. This paper extends the theoretical guarantees for disentanglement to a broader family of contrastive methods, while also relaxing the assumptions about the data distribution. Specifically, we prove identifiability of the true latents for four contrastive losses studied in this paper, without imposing common independence assumptions. The theoretical findings are validated on several benchmark datasets. Finally, practical limitations of these methods are also investigated.

## 1 Introduction

Learning to disentangle the explanatory factors of the observed data is valuable for a variety of machine learning (ML) applications, as it allows for a more compact, interpretable, and manipulable data representation. Consequently, this can improve sampling efficiency [46] and predictive performance [35, 11, 13] for downstream tasks, and promote fairness [33, 8]. Among the many efforts to develop a theoretically grounded approach to learning such representations, contrastive learning (CL) has emerged as a particularly promising technique.

Intuitively, contrastive methods learn a function that maps observations into a representation space, such that related examples (e.g., augmentations of the same sample) are mapped close to each other and negative pairs are mapped far apart [6, 48]. In recent years, this intuition has been refined and extended by different theories. One of these theories is that contrastive methods (approximately) invert the data generating process [49, 24] and thus recover the generative factors. We build upon this line of work and extend their theoretical findings to a wider range of contrastive objectives.

In this study, we aim to develop a unified framework of statistical priors on the data generating process to improve our understanding of CL for disentangled representations. Two important issues we will discuss are how to deal with nonuniform marginal distributions of the latent factors and situations when these factors are conditionally dependent to some degree. We demonstrate how this could be accomplished with a simple modification of common contrastive objectives. We also show that without these modifications, contrastive objectives make implicit assumptions about the data generating process and investigate when these assumptions are justified. We empirically verify our theoretical claims and show practical limitations of the proposed framework.

In short, the contributions of our study can be summarized as follows:

- We extend and unify theoretical guarantees of disentanglement for a family of contrastive losses under relaxed assumptions about the data generating process.

37th Conference on Neural Information Processing Systems (NeurIPS 2023).

- The theoretical findings are empirically validated on several benchmark datasets and we quantitatively compare the disentanglement performance of four contrastive losses.
- We analyze the impact of partially violated assumptions and investigate practical limitations of the proposed framework.

## 2   Related Work

**Disentangled Representation Learning** The concept of learning representations whose components correspond to the explanatory factors of the data can be traced back to the principles of blind source separation [5] and discovering factorial codes [3]. In contrast to earlier work, however, the research focus has gradually shifted to applications where the relationship between data and underlying factors is nonlinear.

The difficulty of this task is that for independent and identically distributed (i.i.d.) data that depend nonlinearly on the latent factors, it is fundamentally impossible to identify these factors without labels or assumptions about the data generating process [23, 34, 26].

A few studies showed how the underlying generative factors can be recovered if they are mutually independent (with at most one of them being Gaussian) and their relation to the observed data exhibits certain regularities, such as a linear mixture followed by an element-wise nonlinearity [42], and local isometries [20]. Gresele et al. [15] also showed that conformal maps rule out some spurious solutions.

Most approaches, however, depart from the i.i.d. assumption and utilize co-observed dependent variables, such as a time index [21] or prior observations [22] in a time-series, observations obtained from interventions [36], data augmentation [47] or different views [14]. These approaches mainly differ in how they model the statistical and causal dependencies between the latent factors and their relation to the auxiliary variable.

Hyvärinen et al. [24] modeled the distribution of the latent factors with the exponential family, where the key assumptions are that, for a given auxiliary variable, the latent factors are conditionally independent and that the effect of the auxiliary variable on their distributions is sufficiently diverse. Their theory was later extended to the Variational Autoencoder (VAE [28]) framework [26] and to energy-based models [27] with further relaxed independence assumptions. Other conditional distributions that have been considered are the Laplace distribution [29] and the von Mises–Fisher distribution [49]. In this work, we extend these results to distance-based conditional distributions and arbitrary marginals of the latents.

Another research direction considers pairs of observations whose underlying latents differ only in some of the factors and share the others, for example, through interventions or actions. In [41, 32] the shared factors or intervention targets are assumed to be known. Locatello et al. [36] showed how the ground-truth latents can be recovered when the pairs of observations share at least one latent factor, which was extended to multi-dimensional factors in [10], but their approaches additionally require that the latent factors are mutually independent. In contrast, [30, 1, 4] proved identifiability for more general causal dependencies between the latent factors by exploiting sparse transitions.

**Contrastive Learning** In recent years, contrastive methods have shown a remarkable ability to learn useful representations for downstream tasks [37, 39, 6, 18, 17]. A number of works attribute this success primarily to their tendency to maximize mutual information (MI) between latent representations [39, 19, 2, 43]. However, it has been observed that contrastive objectives with tighter MI bounds do not necessarily enhance downstream performance [45, 44].

More recently, by exploiting the alignment and uniformity properties [48] of InfoNCE [39], Zimmermann et al. [49] showed that InfoNCE approximately inverts the data generating process, leading to the identification of the true latent factors. We extend this theory to a broader family of contrastive methods [39, 16, 17, 38].

## 3   Contrastive Learning for Disentangled Representations

In this section, we present our theoretical framework of contrastive methods for learning disentangled representations (Figure 1). We begin with the formalization of our assumptions on the underlying data generating process.

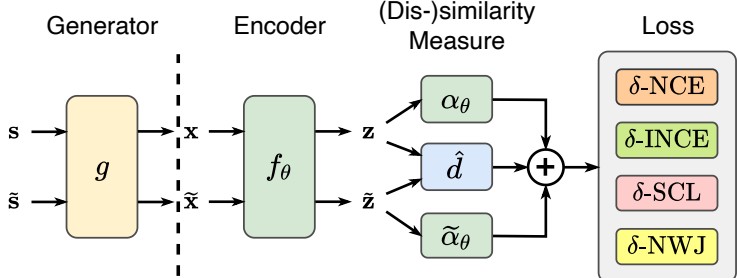

Figure 1: Overview of our CL framework for disentanglement. The unknown generator $g$ maps latent pairs $(\mathbf{s}, \tilde{\mathbf{s}}) \sim p(\mathbf{s}, \tilde{\mathbf{s}})$ to observation pairs $(\mathbf{x}, \tilde{\mathbf{x}})$, which are encoded by $f_\theta$. After computing the dissimilarity measure according to Eq. (2), the representation is optimized with one of the contrastive losses.

### 3.1 Data Generating Process

In this work, we adopt the common assumption that the observations are generated by a smooth and invertible generator $g : \mathcal{S} \to \mathcal{X}$, where $\mathcal{S} \subseteq \mathbb{R}^n$ denotes the space of latent factors and $\mathcal{X} \subseteq \mathbb{R}^m$ the space of observations. Here, we assume $n \le m$. The goal is to find a function $f : \mathcal{X} \to \mathcal{Z} \subseteq \mathbb{R}^n$ that inverts $g$ up to element-wise transformations or other simple mappings such as linear functions.

We do not impose any constraints on the ground-truth marginal distribution of the latents, $p(\mathbf{s})$, except that the support must be connected. Let $\mathbf{s} \sim p(\mathbf{s})$ and $(\mathbf{s}, \tilde{\mathbf{s}})$ be a latent positive pair. We model the dependency between $\mathbf{s}$ and $\tilde{\mathbf{s}}$ with the following conditional distribution

$$p(\tilde{\mathbf{s}}|\mathbf{s}) = \frac{Q(\tilde{\mathbf{s}})}{Z(\mathbf{s})} \, e^{-d(\mathbf{s}, \tilde{\mathbf{s}})} \quad \text{with} \quad Z(\mathbf{s}) = \int_{\mathcal{S}} Q(\tilde{\mathbf{s}}) \, e^{-d(\mathbf{s}, \tilde{\mathbf{s}})} \, d\tilde{\mathbf{s}}, \tag{1}$$

where $Q$ and $Z$ are scalar-valued functions and $d$ is a distance function, which we will specify below. It is worth noticing that here the latent factors $s_i$ are not assumed to be statistically independent or conditionally independent given auxiliary observations ($Q$ cannot necessarily be factorized).

Two related observations (e.g., consecutive frames in a sequence, multiple views or augmentations of the same data point) are then generated by $\mathbf{x} = g(\mathbf{s})$ and $\tilde{\mathbf{x}} = g(\tilde{\mathbf{s}})$. For the sake of simplicity, we assume that the related instances are of the same modality, though it is straightforward to extend this framework to different modalities, such as image and audio coming from different generators.

### 3.2 Contrastive Learning Approaches

Contrastive methods generally have in common that they employ certain mechanisms to measure similarity between data points. The most frequent choice is the cosine similarity [6], while $\ell_p$ distances [7, 49] and even learnable functions [22, 24] have also been used. Inspired by the work in [22, 24], which relies on specific models of learnable similarity measures, we propose the following generic dissimilarity measure

$$\delta(\mathbf{z}, \tilde{\mathbf{z}}) = \hat{d}(\mathbf{z}, \tilde{\mathbf{z}}) + \alpha(\mathbf{z}) + \tilde{\alpha}(\tilde{\mathbf{z}}), \tag{2}$$

where $\alpha$ and $\tilde{\alpha}$ are learnable scalar-valued functions, e.g., neural networks, and $\hat{d}$ is a fixed expression that describes the interaction between related examples. In the ideal case, $\hat{d}$ matches $d$ in Eq. (1).

Let us now recall some common contrastive methods. The first approach we investigate is noise-contrastive estimation (NCE) [16], which has been used for disentanglement in [22, 24]. Using our dissimilarity measure, the loss can be formulated as follows

$$\mathcal{L}_{\delta\text{-NCE}}(f, \delta) = \mathop{\mathbb{E}}_{(\mathbf{x}, \tilde{\mathbf{x}}) \sim p_{\text{pos}}} -\log\left[\text{sig}(-\delta(f(\mathbf{x}), f(\tilde{\mathbf{x}})))\right] - \mathop{\mathbb{E}}_{\mathbf{x}, \mathbf{x}^- \sim p} \log\left[1 - \text{sig}(-\delta(f(\mathbf{x}), f(\mathbf{x}^-)))\right], \tag{3}$$

where sig is the sigmoid function, $p_{\text{pos}}$ is the distribution of positive pairs, and $p$ is the distribution of all observations. [1]

---

[1] For simplicity, we assume here that both marginal distributions are the same. We consider the more general case of different marginals in the supplementary material.

Another popular choice is InfoNCE, which has shown remarkable success in self-supervised learning [39, 6, 19, 18]. The loss function has several related forms and has been referred to by different names [25, 39, 6, 31]. In the following, we use

$$\mathcal{L}_{\delta\text{-INCE}}(f, \delta; K) = \mathop{\mathbb{E}}_{\substack{(\mathbf{x}, \tilde{\mathbf{x}}) \sim p_{\text{pos}} \\ \{\mathbf{x}_i^-\}_{i=1}^K \overset{\text{i.i.d.}}{\sim} p}} - \log \frac{e^{-\delta(f(\mathbf{x}), f(\tilde{\mathbf{x}}))}}{e^{-\delta(f(\mathbf{x}), f(\tilde{\mathbf{x}}))} + \sum_{i=1}^{K} e^{-\delta(f(\mathbf{x}), f(\mathbf{x}_i^-))}}, \tag{4}$$

where $K \geq 1$ is the number of negative examples. Since $\mathcal{L}_{\delta\text{-INCE}}$ is invariant to changes in $\alpha$, we simply set $\alpha(\mathbf{z}) = 0$ in this case.

HaoChen et al. [17] introduced the concept of spectral contrastive learning (SCL), which has classification guarantees for downstream tasks in the context of self-supervised learning. To fit this method into our theoretical framework, we use a slightly modified version in our analysis, where we substitute $f(\mathbf{x})^\mathsf{T} f(\tilde{\mathbf{x}})$ with $e^{-\delta(f(\mathbf{x}), f(\tilde{\mathbf{x}}))}$, i.e.,

$$\mathcal{L}_{\delta\text{-SCL}}(f, \delta) = \mathop{\mathbb{E}}_{(\mathbf{x}, \tilde{\mathbf{x}}) \sim p_{\text{pos}}} -2\, e^{-\delta(f(\mathbf{x}), f(\tilde{\mathbf{x}}))} + \mathop{\mathbb{E}}_{\mathbf{x}, \mathbf{x}^- \sim p} e^{-2\delta(f(\mathbf{x}), f(\mathbf{x}^-))}. \tag{5}$$

Finally, we examine the Nguyen-Wainright-Jordan (NWJ) objective [38] using the same dissimilarity measure as before

$$\mathcal{L}_{\delta\text{-NWJ}}(f, \delta) = \mathop{\mathbb{E}}_{(\mathbf{x}, \tilde{\mathbf{x}}) \sim p_{\text{pos}}} \delta(f(\mathbf{x}), f(\tilde{\mathbf{x}})) + \mathop{\mathbb{E}}_{\mathbf{x}, \mathbf{x}^- \sim p} e^{-\delta(f(\mathbf{x}), f(\mathbf{x}^-))}. \tag{6}$$

Note, that we here use it as a loss instead of a lower bound on MI. To the best of our knowledge, neither the SCL nor the NWJ objective have been employed to learn disentangled representations or for ICA.

### 3.3 Identifiability

In this section, we derive precise conditions on the data generating process under which the relationship between the learned representation and true latent factors can be described by a simple function. In line with [27], we speak of weak identifiability when this function is an affine mapping, and of strong identifiability when it is a generalized permutation matrix.[2]

As in previous research [29, 49], we assume that the interaction between $\mathbf{s}$ and $\tilde{\mathbf{s}}$ is (approximately) known, i.e., $\hat{d}$ in the dissimilarity measure as defined in Eq. (2) matches $d$ in the conditional distribution defined in Eq. (1). Although this is a strong restriction on the latent factors, satisfactory performance is still observed in practice, even when the assumption is violated. Note also that if they only match up to other simple transformations (e.g., invertible element-wise) of the true or learned latents, the recovered relation can be extended by just those transformations.

First, we study the case of weak identifiability. Eq. (1) essentially states that the joint probability density function of positive pairs decreases exponentially with the distance between the underlying latent factors, distorted only by $Q(\tilde{\mathbf{s}})$ and the marginal distribution $p(\mathbf{s})$. Neither $Q(\tilde{\mathbf{s}})$ nor $p(\mathbf{s})$ depends on both instances of a pair. This allows us to learn a distance-preserving representation while accumulating factors that depend exclusively on the first or second element in $\alpha$ and $\tilde{\alpha}$, respectively. We formalize this in the following theorem.

**Theorem 1** (Weak identifiability). *Let $\mathcal{S} \subseteq \mathbb{R}^n$ be open and connected, $\mathcal{X} \subseteq \mathbb{R}^m$, and $g: \mathcal{S} \to \mathcal{X}$ invertible and differentiable. Let us further assume that the observed data satisfy the generative model given in Eq. (1). If $d = \hat{d}$ has one of the following properties:*

*(i) there exists a function $\xi: \mathbb{R}^+ \to \mathbb{R}^+$, such that $\xi \circ d$ is a norm-induced metric[3],*

*(ii) $d(\mathbf{s}, \tilde{\mathbf{s}}) = \sum_i d_i(|s_i - \tilde{s}_i|)$, where each $d_i$ is continuous and strictly increasing,*

*then the optimal estimator of any of the contrastive losses presented above identifies the true latent factors up to affine transformations, i.e., $h = f \circ g$ is an affine mapping.*

---

[2]This property is sometimes called explicitness [40] or informativeness [9], when it is left open what is meant by a simple function.

[3]A norm-induced metric is a metric which is translation invariant, $d(\mathbf{x}, \mathbf{y}) = d(\mathbf{x} + \mathbf{b}, \mathbf{y} + \mathbf{b})$, and absolutely homogeneous, $d(\sigma\mathbf{x}, \sigma\mathbf{y}) = |\sigma| d(\mathbf{x}, \mathbf{y})$

The central idea of the proof, which can be found in the supplementary material, is to show that the learned representation preserves the distance from the latent space, i.e., $d(\mathbf{s}, \tilde{\mathbf{s}}) = d(h(\mathbf{s}), h(\tilde{\mathbf{s}}))$. Further, by constraining the dimension and connectivity of the latent space, such a mapping must be affine. As a byproduct of the derivation, we obtain for the global optimum that

$$\tilde{\alpha} \circ h = \log p - \log Q \qquad (7)$$

and, except for $\mathcal{L}_{\delta\text{-INCE}}(f, \delta; K)$,

$$\alpha \circ h = \log Z. \qquad (8)$$

We verify this with a simple example in Figure 2. This means that when we optimise the InfoNCE objective with $\tilde{\alpha}$ set to zero, we implicitly assume that $Q = p$. This is true, for example, when the marginal distribution is uniform and $Q$ is constant. Thus, our theorem is consistent with the perspective of uniformity and alignment [48] and can be seen as a generalization of Theorem 5 from [49]. In particular, we extend their results to other contrastive losses, nonconvex latent spaces, nonuniform marginal distributions, and even to the case where the latent factors are not conditionally independent.

Next we show strong identifiability when $d$ in the conditional distribution of the latent factors is further restricted. The proof can be found in the supplementary material.

**Theorem 2** (Strong identifiability). *Assume that all conditions in Theorem 1 are satisfied. Let the function $d$ from Eq. (1) be defined by*

$$d(\mathbf{s}, \tilde{\mathbf{s}}) = \sum_i (|s_i - \tilde{s}_i|/\sigma_i)^\beta, \qquad (9)$$

*with $\beta \in (0, 2) \cup (2, \infty)$ and $\sigma_i > 0$ for all $i$, then $h = f \circ g$ is a generalized permutation matrix, i.e., a composition of a permutation and element-wise scaling and sign flips.*

This result is similar to earlier theorems [29, 49], but relaxes some conditions. In particular, it also holds for nonuniform marginals and $\beta \in (0, 1)$. For the case $\beta = 2$, the conditional distribution is quasi-Gaussian [22] and, as with linear Independent Component Analysis (ICA) for normally distributed sources, strong identifiability is then provably impossible.

In contrast to [24], we show identifiability for the generalized normal distribution (excluding $\beta = 2$) and do not assume some regularity conditions, e.g., that the encoder is twice differentiable and invertible. However, we do not claim to generalize their results, since the exponential family contains distributions that we do not consider here.

Theorem 2 also has an interesting connection to some recently proposed disentanglement methods that rely on sparsity [30, 1]. This situation is similar to the case when $\beta \to 0$ in Eq. (9) and the conditional $p(\tilde{\mathbf{s}}|\mathbf{s})$ becomes sparse. However, this is a topic for future work.

## 4 Experiments

In this section, we conduct a comprehensive quantitative evaluation of our theoretical framework. We largely adopt the experimental setup and evaluation protocol of [49] with one major difference. During training, we additionally optimize for $\alpha$ and $\tilde{\alpha}$ in Eq. (2), both of which are parameterized by three-layer neural networks, respectively. In practice, both functions are normalized to a mean of zero and are jointly trained with a learnable offset $c$, i.e., $\delta(\mathbf{z}, \tilde{\mathbf{z}}) = \hat{d}(\mathbf{z}, \tilde{\mathbf{z}}) + \alpha(\mathbf{z}) + \tilde{\alpha}(\tilde{\mathbf{z}}) + c$. In the case of $\delta$-INCE, we use $\delta(\mathbf{z}, \tilde{\mathbf{z}}) = \hat{d}(\mathbf{z}, \tilde{\mathbf{z}}) + \tilde{\alpha}(\tilde{\mathbf{z}})$ instead. Further details on the implementation can be found in the supplementary material.

As in previous research [21, 22, 49], we test for weak identifiability, i.e., up to affine transformations, by fitting a linear regression model between the ground-truth and recovered latents and compute the coefficient of determination ($R^2$). For strong identifiability, i.e., up to generalized permutations and element-wise transformations, we report the mean correlation coefficient (MCC).

### 4.1 Validation of Theoretical Findings

To validate our theoretical claims, we adopt a generative process similar to [49]. However, to demonstrate that identifiability can be achieved for nonconvex latent spaces, nonuniform marginals,

Table 1: Identifiability on synthetic data. MCC [%] mean $\pm$ standard deviation over 2 random seeds

| Scenario | $\beta$ | $\delta$-NCE | $\delta$-INCE | $\delta$-SCL | $\delta$-NWJ |
|---|---|---|---|---|---|
| Box (simple) | 1/2 | $99.68 \pm 0.04$ | $99.56 \pm 0.18$ | $87.08 \pm 1.40$ | $99.80 \pm 0.02$ |
| Box (simple) | 1 | $99.91 \pm 0.01$ | $99.90 \pm 0.01$ | $94.33 \pm 2.02$ | $99.87 \pm 0.01$ |
| Box (simple) | 3 | $99.66 \pm 0.20$ | $96.97 \pm 2.47$ | $98.29 \pm 0.14$ | $99.71 \pm 0.05$ |
| Box (simple) | 5 | $99.79 \pm 0.02$ | $96.56 \pm 2.44$ | $98.63 \pm 0.36$ | $99.74 \pm 0.00$ |
| Box (complex) | 1 | $99.13 \pm 0.41$ | $99.87 \pm 0.05$ | $83.73 \pm 2.49$ | $93.14 \pm 4.97$ |
| Box (complex) | 3 | $99.90 \pm 0.00$ | $99.84 \pm 0.00$ | $93.64 \pm 0.22$ | $99.84 \pm 0.00$ |
| Hollow ball | 1 | $99.73 \pm 0.11$ | $99.70 \pm 0.08$ | $90.27 \pm 6.48$ | $99.73 \pm 0.08$ |
| Hollow ball | 3 | $95.82 \pm 4.65$ | $98.05 \pm 0.08$ | $97.65 \pm 0.00$ | $99.03 \pm 0.06$ |
| Hollow ball | 5 | $98.52 \pm 0.21$ | $96.25 \pm 0.43$ | $97.57 \pm 0.06$ | $98.67 \pm 0.16$ |
| Cube grid | 1 | $99.87 \pm 0.02$ | $99.66 \pm 0.04$ | $96.92 \pm 1.88$ | $99.79 \pm 0.01$ |
| Cube grid | 5 | $97.42 \pm 0.12$ | $82.26 \pm 7.06$ | $96.17 \pm 0.07$ | $97.96 \pm 0.06$ |

and without conditionally independent latents, we investigate additional configurations of these components.

In this section, we consider latent spaces with $n = 10$ dimensions. We generate pairs of source signals $(\mathbf{s}, \tilde{\mathbf{s}})$ by first sampling from a marginal distribution $p(\mathbf{s})$ and subsequently from a conditional distribution

$$p(\tilde{\mathbf{s}}|\mathbf{s}) = \frac{Q(\tilde{\mathbf{s}})}{Z(\mathbf{s})} e^{- \sum_i \left( \frac{|s_i - \tilde{s}_i|}{\sigma} \right)^\beta} . \tag{10}$$

The observations $(\mathbf{x}, \tilde{\mathbf{x}})$ are then generated by passing $\mathbf{s}$ and $\tilde{\mathbf{s}}$ individually through an (approximately) invertible multilayer perceptron (MLP).

We analyze the following four scenarios where we vary $\mathcal{S}$, $p(\mathbf{s})$, $Q$ and $\beta$:

- Box (simple): $\mathcal{S} = [0, 1]^n$, $p(\mathbf{s})$ is uniform, $Q$ is constant. This is our baseline.

- Box (complex): $\mathcal{S} = [0, 1]^n$, $p(\mathbf{s})$ is a normal distribution with block-diagonal correlation matrix such that the odd dimensions are correlated with the even dimensions, $Q$ has a checkerboard pattern (i.e., has value 1 on "white squares" and 0.1 on "black squares"), so the latent factors are not conditionally independent.

- Hollow ball: $\mathcal{S} = \{\mathbf{s} \in \mathbb{R}^n \mid r < \|\mathbf{s}\| < R, 0 < r < R\}$ with inner radius $r$ and outer radius $R$, $p(\mathbf{s})$ is uniform over the radius and angle (but not uniform in Euclidean coordinates), $Q$ is constant. Note that here $\tilde{s}_i$ and $\tilde{s}_j$ are not conditionally independent given $\mathbf{s}$, because the support is not rectangular.

- Cube grid: $\mathcal{S} = \{\mathbf{s} \in [-1, 1]^n \mid \forall i : s_i < -b \vee s_i > b, 0 < b < 1\}$ (i.e., $\mathcal{S}$ is disconnected), $p(\mathbf{s})$ is uniform, $Q$ is constant. Although our current analysis makes no predictions about disconnected spaces, we investigate this configuration to empirically probe potential limits of our framework.

To fit our hardware setup, we use a smaller batch size of 5120. Also, we use an encoder network with residual connections and batch normalization, which has been shown to be more stable. We generally leave the output space unrestricted, i.e., $\mathcal{Z} = \mathbb{R}^n$.

Table 1 provides a summary of the MCC scores. The corresponding $R^2$ values can be found in the supplementary material. In each scenario, all contrastive losses achieve near-optimal disentanglement, with the exception of a few outliers. In particular, $\delta$-SCL appears to be numerically less stable than its counterparts, which we will examine in more detail in later sections. Additional experiments in the supplementary material compare $\delta$-SCL with the original SCL and show that $\delta$-SCL generally yields higher disentanglement scores. However, we would like to point out that one should not draw direct conclusions from these results about the performance of the original SCL on other downstream tasks.

To verify that the learned functions $\tilde{\alpha}$ and $\alpha$ converge to the solution given in Eq. (7) and Eq. (8), we construct a simple example where the ground truth is known. We choose $\mathcal{S} = \mathcal{Z} = [0, 1]^2$ with a uniform marginal distribution. The positive examples are sampled from a truncated Laplace

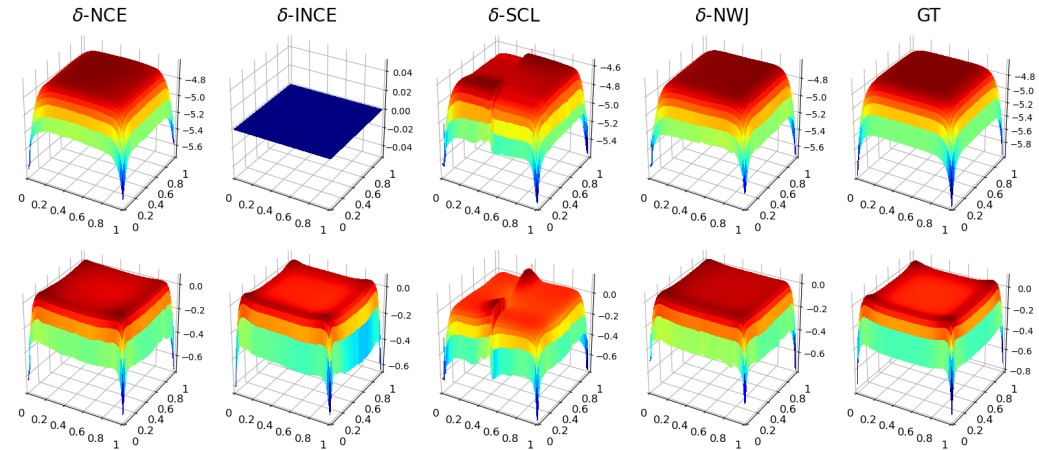

Figure 2: Learned functions $\alpha$ (top row) and $\tilde{\alpha}$ (bottom row) for a simple example described in the text. The ground truth is in the right column. In the case of $\delta$-INCE, $\alpha$ is set to zero.

Table 2: MCC [%] scores on synthetic data for $\alpha(\mathbf{z}) = \tilde{\alpha}(\tilde{\mathbf{z}}) = c$

| Scenario | $\beta$ | $\delta$-NCE | $\delta$-INCE | $\delta$-SCL | $\delta$-NWJ |
|---|---|---|---|---|---|
| Box (simple) | 1/2 | 99.57 | 99.81 | 82.69 | 89.22 |
| Box (simple) | 1 | 99.85 | 99.91 | 94.00 | 99.81 |
| Box (simple) | 3 | 99.77 | 99.40 | 99.78 | 99.76 |
| Box (simple) | 5 | 99.82 | 98.86 | 94.84 | 99.82 |
| Box (complex) | 1 | 99.70 | 99.82 | 76.63 | 99.68 |
| Box (complex) | 3 | 99.69 | 99.75 | 89.78 | 99.51 |
| Hollow ball | 1 | 98.49 | 99.37 | 87.31 | 97.94 |
| Hollow ball | 3 | 98.59 | 97.23 | 96.39 | 98.64 |
| Hollow ball | 5 | 98.52 | 96.26 | 98.49 | 98.52 |
| Cube grid | 1 | 99.82 | 99.39 | 98.21 | 99.75 |
| Cube grid | 5 | 96.53 | 84.73 | 97.50 | 97.42 |

distribution. Figure 2 shows the results. Details and additional experimental setups can be found in the supplementary material.

Since we learn the sum of $\alpha$ and $\tilde{\alpha}$, they can only be identified up to some offset. In Figure 2, we have added the learned bias to $\alpha$ to confirm that the total offset is correct. All losses reach the global optimum except for $\delta$-SCL, which appears to be stuck at a local minimum.

### 4.2 Evaluation under Partially Violated Assumptions

We also report results for the case where $\alpha$ and $\tilde{\alpha}$ are set to a learnable constant (see Table 2). Interestingly, we do not observe that this change significantly affects the outcome. However, we do observe a drastic performance drop in $\delta$-NCE, $\delta$-SCL and $\delta$-NWJ when this offset is fixed to a value which is far from the ground truth. Specifically, setting the bias to zero causes a drop in the MCC score by more than 20% on average when $\beta = 1$. For larger $\beta$ we do not observe different behaviour. This is likely due to the fact that the optimal bias is far from 0 when $\beta$ is smaller, which can distort the learned representation.

### 4.3 Evaluation on Kitti Masks

The KITTI Masks dataset [29] consists of segmentation masks of pedestrians extracted from the autonomous driving benchmark KITTI-MOTS [12]. We compare the presented loss functions with

Table 3: KITTI Masks. MCC [%] mean $\pm$ standard deviation over 10 random seeds

| $\overline{\Delta t}$ | Loss | Laplace | | Normal | |
| --- | --- | --- | --- | --- | --- |
| | | Unbounded | Box | Unbounded | Box |
| 0.05s | SlowVAE [29] | $66.1 \pm 4.5$ | | | |
| $-"-$ | $\delta$-contr [49] | $77.1 \pm 1.0$ | $74.1 \pm 4.4$ | $58.3 \pm 5.4$ | $59.9 \pm 5.5$ |
| $-"-$ | $\delta$-INCE | $77.3 \pm 1.5$ | $75.9 \pm 1.7$ | $68.1 \pm 5.0$ | $68.3 \pm 4.8$ |
| $-"-$ | $\delta$-NCE | $77.1 \pm 1.0$ | $75.4 \pm 2.4$ | $60.2 \pm 4.9$ | $68.0 \pm 1.5$ |
| $-"-$ | $\delta$-SCL | $75.8 \pm 3.9$ | $71.5 \pm 7.5$ | $66.2 \pm 10.7$ | $66.5 \pm 4.0$ |
| $-"-$ | $\delta$-NWJ | $77.3 \pm 0.7$ | $77.7 \pm 0.9$ | $67.5 \pm 5.3$ | $69.2 \pm 5.3$ |
| 0.15s | SlowVAE [29] | $79.6 \pm 5.8$ | | | |
| $-"-$ | $\delta$-contr [49] | $79.4 \pm 1.9$ | $80.9 \pm 3.8$ | $60.2 \pm 8.7$ | $68.4 \pm 6.7$ |
| $-"-$ | $\delta$-INCE | $80.5 \pm 1.4$ | $77.0 \pm 3.3$ | $68.9 \pm 5.2$ | $68.4 \pm 4.2$ |
| $-"-$ | $\delta$-NCE | $79.3 \pm 2.3$ | $79.5 \pm 2.7$ | $62.7 \pm 4.6$ | $68.4 \pm 3.5$ |
| $-"-$ | $\delta$-SCL | $72.9 \pm 3.9$ | $74.4 \pm 2.6$ | $62.5 \pm 5.0$ | $66.1 \pm 5.7$ |
| $-"-$ | $\delta$-NWJ | $79.3 \pm 2.0$ | $78.2 \pm 6.1$ | $68.2 \pm 3.5$ | $72.6 \pm 6.7$ |

Table 4: 3DIdent. $R^2$ [%] and MCC [%] mean $\pm$ standard deviation over 2 random seeds

| $\beta$ | Loss | $R^2$ | | MCC | |
| --- | --- | --- | --- | --- | --- |
| | | Unbounded | Box | Unbounded | Box |
| 2 | $\delta$-contr [49] | $96.43 \pm 0.03$ | $96.73 \pm 0.10$ | $54.94 \pm 0.02$ | $98.31 \pm 0.04$ |
| 2 | $\delta$-INCE | $97.87 \pm 0.07$ | $98.08 \pm 0.03$ | $56.09 \pm 0.10$ | $98.98 \pm 0.09$ |
| 2 | $\delta$-NCE | $98.05 \pm 0.04$ | $97.91 \pm 0.02$ | $55.34 \pm 0.11$ | $98.95 \pm 0.08$ |
| 2 | $\delta$-SCL | $71.07 \pm 3.76$ | $74.92 \pm 2.28$ | $49.88 \pm 1.20$ | $76.02 \pm 3.13$ |
| 2 | $\delta$-NWJ | $94.03 \pm 1.35$ | $97.67 \pm 0.02$ | $53.82 \pm 0.46$ | $98.79 \pm 0.02$ |
| 1 | $\delta$-contr [49] | | $96.87 \pm 0.08$ | | $98.38 \pm 0.03$ |
| 1 | $\delta$-INCE | $98.03 \pm 0.06$ | $97.45 \pm 0.04$ | $55.79 \pm 0.09$ | $98.63 \pm 0.05$ |
| 1 | $\delta$-NCE | $97.83 \pm 0.04$ | $97.33 \pm 0.03$ | $56.35 \pm 0.08$ | $98.71 \pm 0.02$ |
| 1 | $\delta$-SCL | $78.59 \pm 3.92$ | $93.13 \pm 1.83$ | $52.28 \pm 1.98$ | $95.77 \pm 2.58$ |
| 1 | $\delta$-NWJ | $94.02 \pm 1.28$ | $97.64 \pm 0.09$ | $54.87 \pm 0.33$ | $98.82 \pm 0.03$ |

the current state of the art for this dataset [29, 49]. Except for the additional parameters in $\delta$, we use the same settings and follow their evaluation protocol.

As noted by Klindt et al. [29], the transitions of the latents are mostly sparse. It is therefore not surprising that all contrastive losses consistently perform better when a Laplace conditional is assumed as opposed to a normal conditional. Considering the large variance across different random seeds, all contrastive losses perform similarly well, with the largest outlier being $\delta$-SCL, which could already be observed on the synthetic dataset in the previous section.

Overall, there are only small differences between the different embedding spaces, with about half of the methods performing better on each setup. Consistent with previous observations [49], we see for all models slightly better performance for larger time intervals, $\overline{\Delta t}$, between consecutive frames. One possible reason is that a larger concentration of the conditional leads to a larger range of $\delta$ values, causing numerical instability. We investigate this phenomenon in more detail in Section 4.5.

### 4.4 Evaluation on 3DIdent

We additionally evaluate our framework under partially violated assumptions on 3DIdent [49], a recently proposed benchmark dataset. The dataset is composed of rendered images showing a complex object in different positions, rotations and under different lighting conditions. Positive pairs $(\mathbf{s}, \tilde{\mathbf{s}})$ are obtained by randomly selecting $\mathbf{s}$ and matching sampled $\tilde{\mathbf{s}}'$ from the conditional $p(\tilde{\mathbf{s}}'|\mathbf{s})$ to the closest $\tilde{\mathbf{s}}$ with a rendered image. We consider a normal ($\beta = 2$) and a Laplace conditional ($\beta = 1$). In both cases we set $\hat{d}(\mathbf{z}, \tilde{\mathbf{z}}) = \|\mathbf{z} - \tilde{\mathbf{z}}\|_2^2$ where we either leave the output space unbounded ($\mathcal{Z} = \mathbb{R}^n$) or restrict it to an adjustable box ($\mathcal{Z} = [0, b]^n$, where $b$ is a learnable parameter). The latter

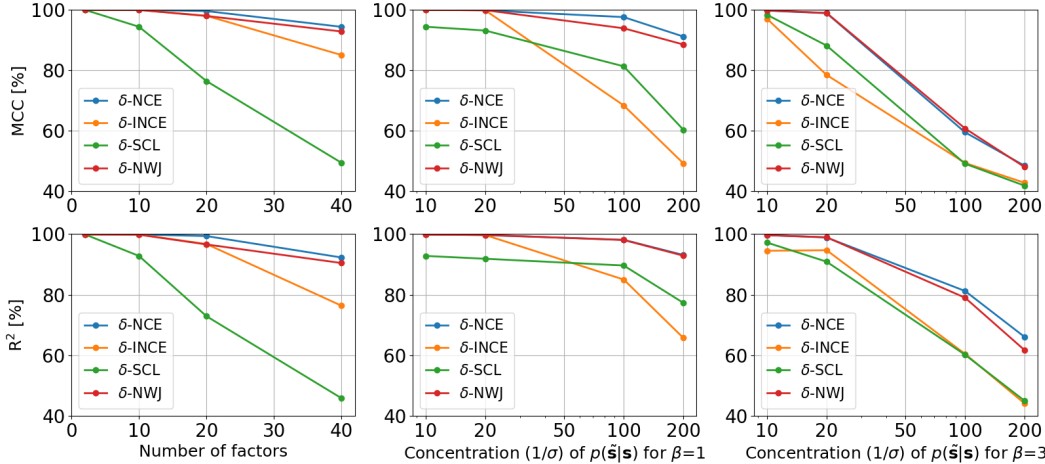

Figure 3: MCC (top) and $R^2$ scores (bottom) for different numbers of latent factors (left) and for different concentrations $(1/\sigma)$ of the conditional distribution for $\beta = 1$ (center) and $\beta = 3$ (right).

is achieved by applying a sigmoid function in the last layer of the network and multiplying the result by $b$. Table 4 shows the results for weak and strong identifiability.

With the exception of $\delta$-SCL, the examined contrastive losses have $R^2$ values close to the theoretical optimum. However, $\delta$-NWJ shows less robustness when we do not restrict the latent space compared to $\delta$-NCE and $\delta$-INCE. This demonstrates that incorporating knowledge about the latent space can stabilize training and lead to strong identifiability, even in the case of a (truncated) Gaussian conditional distribution.

## 4.5 Investigation of Limitations

In this section, we discuss practical and theoretical limitations of the proposed framework. We first examine how changes in the number of latent factors and the concentration of the conditional distribution affect disentanglement. We use the same setup as in Section 4.1, except that we choose a slightly smaller batch size of 4096 and train for $8 \times 10^5$ iterations to ensure convergence when we consider more dimensions. The results are shown in Figure 3.

First, we note that all methods have difficulties when the number of latent factors is increased (see Figure 3, left column). In particular, $\delta$-SCL shows little robustness for larger dimensions.

Furthermore, we see that the performance of all methods declines with increasing concentration $(1/\sigma)$ of the conditional and is exacerbated by a larger shape parameter $(\beta)$. The work in [49] noted that InfoNCE has difficulty disentangling the latent factors when the conditional distribution becomes too flat relative to the marginal one, which, they pointed out, makes positive examples indistinguishable from negative ones. We observe here the same problem in the opposite case.

Consider the following example. Suppose $\mathcal{S} = [0, 1]^n$ and $p(\mathbf{s})$ is uniform, then the range of $\delta$ at the global optimum is $[0, n/\sigma^\beta]$. For larger $\beta$ and smaller $\sigma$, this quickly leads to vanishing values when calculating the exponential in the contrastive losses. This could also be one of the reasons why we see worse performance for smaller $\overline{\Delta t}$ in KITTI Masks.

It should be noted that employing learnable functions in the similarity measure may entail certain drawbacks, such as the need to store additional parameters and a slight increase in training time. Furthermore, we have seen in Section 4.2 that in some cases the underlying data distribution can be adequately approximated by constant $\alpha$ and $\tilde{\alpha}$. Although not observed in this work, it is also plausible that the inclusion of these functions for more complex datasets may increase the risk of getting stuck in poor local minima during training, as incorrect shapes of these functions may promote poor representations (the shape of $\alpha$ and $\tilde{\alpha}$ affects the gradient of $f$ and vice versa).

Finally, we want to emphasize that the mechanism that governs the observation pairs is highly dependent on the application and that the relationship between the latents may not be adequately

described by a distance-based conditional as assumed in this work (Eq. (1)). However, we leave the exploration of methods suitable for other mechanisms to future work.

## 5 Conclusion

In this work, we propose a new theoretical framework of contrastive methods for learning disentangled representations. In contrast to previous research, our framework accounts for nonuniform marginal distributions of the factors of variation, allows for nonconvex latent spaces, and does not assume that these factors are statistically independent or conditionally independent given an auxiliary variable.

Our theory unveils that contrastive methods implicitly encode assumptions about the data generating process. We show empirically that even when these assumptions are partially violated, these methods learn to recover the true latent factors. This study provides further evidence that contrastive methods learn to approximately invert the underlying generative process, which may explain their remarkable success in many applications.

## 6 Acknowledgements

This work has been carried out as part of the AuSeSol-AI Project which is funded by Federal Ministry for the Environment, Nature Conservation, Nuclear Safety and Consumer Protection (BMUV) under the Grant 67KI21007F.

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
