# OpenReview forum: "Towards a Unified Framework of Contrastive Learning for Disentangled Representations"
_NeurIPS.cc/2023/Conference — NeurIPS 2023 poster_

### Official Review · Reviewer_bVRB · 2023-07-06

**Soundness:** 3 good
**Presentation:** 3 good
**Contribution:** 2 fair
**Rating:** 5
**Confidence:** 2

**Summary:**

In this paper the authors propose a new dissimilarity measure function for contrastive representation learning. They provide the identifiability of the factors of variations in four different contrastive objectives. Specifically, they describe the connection between the learned representations and true factors using some function in the data generating process. They extend the property to cases where there are non-convex latent spaces or/and non-uniform marginal distributions. Empirically they validate their approach on several datasets.

**Strengths:**

This paper is clearly written; The authors make it clear what the position of their approach in the literature of contrastive learning. The similarity measure function is simple to implement. I find it useful to discuss the strong and weak identifiability under different conditions, and the derivation is mathematically sound in my opinion. Though contrastive learning have been studies for a few years, it is interesting to investigate and unify the theoretical guarantees of disentanglement for a family of contrastive objectives under different assumptions.

**Weaknesses:**

1. I am a bit confused about the problem definition. While the authors try to model the relationship between true factors and the learned representations, in most cases we don't really know the true factor. Instead what we typically do in contrastive learning is to measure the similarity between pairs of observations where there are slight differences from certain perspectives, i.e. one image is the rotated version of the other. So I would hope that the authors can talk a bit more about how the identifiability works in practical scenarios.

2. I wonder whether the authors need to any data augmentations when training with the contrastive loss. There does not seem to be any discussion on that.


**Questions:**

Please see my questions in the previous section.

**Limitations:**

I think there is no potential negative societal impact of their work.

---

> ### Author Rebuttal · Authors · 2023-08-08
>
> We appreciate that our contribution is valued, and we are grateful for the comments on where we can provide further elaboration.
>
> **Identifiability in practical scenarios**
>
> The scenario described by the reviewer comes closest to our experiments with 3DIdent. However, instead of using data augmentation, the positive pairs can be considered as coming from successive frames of a video (same for KITTI Masks) where the underlying factors (e.g., position, rotation) change with some probability. Data augmentation behaves similarly, but does not allow, for example, to rotate or translate an object in all directions.
> Our theory assumes that such a mechanism for creating positive pairs changes the latent factors according to Eq. 1. This equation says that the factors we want to disentangle must at least sometimes change. This property is fundamental in disentanglement theory and is sometimes referred to as variability. It is important to note that it generally does not apply to pure data augmentation. To our knowledge, separating changing from non-changing factors is the best that can be provably achieved without supervision in the case where some factors never change [1].
>
> **Do we use data augmentation?**
>
> We do not use data augmentation in this study. Rather, we leave it open how positive pairs are chosen for the contrastive loss, as this is highly dependent on the application and data augmentation may not be an option or the only option.
>
> **References**
>
> [1] Self-Supervised Learning with Data Augmentations Provably Isolates Content from Style

---

> > ### Comment · Area_Chair_W6wf · 2023-08-18
> >
> > Dear Authors,
> >
> > The reviewer did not acknowledge your response, so I am joining the discussion. I have read the review and rebuttal and have no further questions.
> >
> > Kind regards,
> > Your AC

---

### Official Review · Reviewer_m4mq · 2023-07-06

**Soundness:** 4 excellent
**Presentation:** 3 good
**Contribution:** 3 good
**Rating:** 6
**Confidence:** 4

**Summary:**

This paper presents an analysis focusing on the disentanglement of the characteristics of internal representations obtained by contrastive learning. The aim is to recover the internal representation of the true latent space up to some transformation by placing certain assumptions on the data generation process. Particularly, it generalizes Zimmerman's theoretical analysis so that it can be applied to some loss function variants and relaxes the data distribution assumptions. The paper successfully derives the results of identifiability without the common assumption of independence in the research of disentangled representation.

**Strengths:**

While I could not fully follow the entire proof, the theoretical claim appears technically sound and solid. Including the analysis of the identifiability of Nonlinear ICA, the theoretical contributions obtained in this study are meaningful for the community focusing on acquiring disentangled representation with theoretical guarantees. Significantly, they have removed the independence assumption typically used in the analysis of disentangled representation. As it is difficult to sample all disentanglement factors in real-world datasets uniformly, this is an important direction for constructing practical theories. However, there are already several preceding studies on the approach of removing the independence assumption, so the positioning of this paper should be properly discussed, which I will mention in the Weakness section.

**Weaknesses:**

**There are certain concerns about the gap between theoretical claims and realistic experimental settings.** The fundamental theoretical claims in this paper are based on the conditional distribution in the true latent variable space defined in Equation 1. As far as I understand the claims in the paper, the identifiability can be guaranteed by training with the contrastive learning objective by assuming that the positive pair samples can be obtained in the form following this latent conditional distribution. However, it is questionable how much validity can be claimed to assume that positive pair samples can be obtained in such a distribution form when considering practical situations. If the true joint distribution of positive pairs is considered known, supervised learning should be done by simply generating labels from the distribution, so the significance should be in the setting where samples can be implicitly obtained while the explicit form of distribution is unknown. This concern relates to the practical applicability of the theoretical claims concluded in the paper and is important in positioning the contributions and significance of this paper. For example, SimCLR, widely known for its practical effectiveness, uses two different views augmented from data as a positive pair, but can this be justified from the perspective of Equation 1? How meaningful is it to consider the corresponding latent variable space?

**It is necessary to show the contributions of existing related studies appropriately.** While the paper claims to have successfully derived the results of identifiability without the typical independence assumption in the research of disentangled representation, there are already preceding studies that theoretically guarantee disentanglement without imposing independence assumption using weak supervision [1, 2]. In particular, while [1] adopts a GAN-based approach for training the model, it strongly relates to the problem setting of this study, as it uses pairs that share a part of the disentangled factors as supervision signals. What advantages can be recognized in the preceding studies compared to these approaches?

**A fair comparison is needed between contrastive objectives.** In the paper, a slightly different objective function is adopted from the original spectral contrastive learning (SCL) objective in order to fit into the theoretical framework of the proposed method (Equation 5). Some numerical experiments have pointed out the numerical instability of SCL, but could this be due to the influence of modifying the original objective function? It should be verified whether the same problem occurs in the objective function presented in the original paper. If the numerical instability pointed out in this paper is due to the modification introduced, it should be explicitly mentioned, as it could potentially mislead readers about the effectiveness of SCL.

**References**

1. Shu, R., Chen, Y., Kumar, A., Ermon, S., & Poole, B. (2019, September). Weakly Supervised Disentanglement with Guarantees. In *International Conference on Learning Representations*.
2. Locatello, F., Poole, B., Rätsch, G., Schölkopf, B., Bachem, O., & Tschannen, M. (2020, November). Weakly-supervised disentanglement without compromises. In *International Conference on Machine Learning* (pp. 6348-6359). PMLR.

**Questions:**

- **In the numerical experiments using synthetic datasets, why were the values of $\beta$ not set to be compositional for each scenario?** For example, $\beta=1/2$ tested in the Box (simple) scenario does not seem to be tested in other scenarios like Hollow ball and Cube grid. Specifically, for the Cube grid scenario, only this condition seems to skip the intermediate value of $\beta=3$. Is there a clear reason for this? The validity of the parameter range used in the experiments should be clearly indicated.
- **Sampling of positive pairs seems to be a crucial assumption in theory, but I could not glean from the text what assumptions were made.** In the Weakness section above, I have raise some concerns based on my understanding that positive pairs are sampled from a probability distribution derived from the conditional distribution of the true latent variables defined in Equation 1. Is this understanding correct? If this interpretation is wrong, it would be helpful to explicitly mention in the text about the assumption of the probability distribution of the positive pair samples.

**Limitations:**

This paper analyzes the acquisition of disentangled representation using contrastive learning from the viewpoint of identifiability, successfully relaxing some of the assumptions required in previous research. This is a theoretically solid result, and the results have been verified in numerical experiments using several benchmark datasets. However, as mentioned at the beginning of the Weakness section, there are significant concerns about applying the claims of this paper to practical situations. By appropriately discussing the practical validity of the assumed premises, I believe the significance of this paper will become more firmly established.

---

> ### Author Rebuttal · Authors · 2023-08-08
>
> We thank the reviewer for the detailed assessment and for pointing out where further comparison and discussion may be needed.
>
> **Gap between theoretical claims and realistic experimental settings**
>
> It is true that the assumed conditional (Eq. 1) is not fully consistent with data augmentation, and it is useful to point this out in the paper, but this is far from the only practical application.
> We would also like to point out that there is a difference between knowing the distribution of observed pairs and latent pairs. The latter does not allow labels to be generated and is sometimes even partially known. This may motivate, for example, a partial labeling.
>
> **Contributions of existing related studies**
>
> We will add a discussion on weakly supervised disentanglement methods in the revised version. Both references mentioned have similarities with the limit case of Eq. 1, when $\beta \to 0$ leads to an l0-"norm" or sparsity prior (i.e., only some latent factors differ in a positive pair). Unlike [1, 2], Eq. 1 also allows for dense transitions, but in the sparse case our approach may not be as numerically stable as methods designed specifically for this case. We think further research is needed to fully understand the theoretical relationships and practical issues.
>
> **A fair comparison is needed between contrastive objectives**
>
> To address this point we have conducted further experiments with the original SCL loss. Please see the general response for details.
>
> **Why not compositional $\beta$ values?**
>
> We fully agree and understand the reviewer's concerns. The set of chosen $\beta$ values were mainly due to computational constraints available for preparing the submission. Our experiments suggest that convergence in terms of elapsed time can vary significantly, depending on the support of data, distributions, loss functions, hyper-parameters, etc. Therefore, in order to draw conclusive statements from the experiments to support the theoretical findings with a fixed number of iterations, e.g. 3e5, we only reported the results with almost converged experiments, which we believe to be sufficient to support the theoretical claim about the identifiability of nonlinear disentangled representations.
>
> In the last week, we ran additional experiments for different scenarios. Here are the results after 3e5 iterations for “cube grid” with $\beta=3$:
>
> MCC: $\delta$-NCE: 98.62 | $\delta$-INCE: 68.33 | $\delta$-SCL: 93.27 | $\delta$-NWJ: 98.55
>
> R2:  $\delta$-NCE: 98.34 | $\delta$-INCE: 92.07 | $\delta$-SCL: 93.06 | $\delta$-NWJ: 98.33
>
> and “box (complex)” with $\beta=5$:
>
> MCC: $\delta$-NCE: 92.08 | $\delta$-INCE: 99.92 | $\delta$-SCL: 92.90 | $\delta$-NWJ: 99.91
>
> R2: $\delta$-NCE: 98.21 | $\delta$-INCE: 99.87 | $\delta$-SCL: 97.34 | $\delta$-NWJ: 99.89
>
> In the cases where the scores are a little lower, the optimization has not yet completely converged. To illustrate this, we have included a figure showing the convergence of $\delta$-INCE for various $\beta$ values on “cube grid”. Also the experiments with $\beta<1$ take much longer than $\beta>=1$ due to division by zero checks when computing the gradients.
>
> **Sampling positive pairs**
>
> The understanding is correct. In the revised version, we will make the sampling procedure clearer.

---

> > ### Comment · Area_Chair_W6wf · 2023-08-18
> >
> > Dear Authors,
> >
> > The reviewer did not acknowledge your response, so I am joining the discussion. I have read the review and rebuttal and have no further questions.
> >
> > Kind regards,
> > Your AC

---

> > ### Comment · Reviewer_m4mq · 2023-08-22
> > **Response**
> >
> > Thank you for responding to my questions.
> >
> > **Gap between theoretical claims and realistic experimental settings**
> >
> > > We would also like to point out that there is a difference between knowing the distribution of observed pairs and latent pairs. The latter does not allow labels to be generated and is sometimes even partially known. This may motivate, for example, a partial labeling.
> >
> > I see that the authors and I agree on the differences between the former and the latter. While the partial labels mentioned by the authors may indeed be one of the practical assumptions, I still have slight concerns about the applicability of the authors' proposed problem setup as it is. In any case, I believe that the problem setting of this analysis should be clearly stated in the manuscript, specifically what practical situation is envisioned and the limits of its applicability.
> >
> > **Sampling positive pairs**
> >
> > Based on the authors' individual & general responses, I found that my initial understanding was correct. If so, my first concern still remains. I believe that we cannot say that the assumption that data can be sampled in IID from a probability distribution conditioned on a *true latent variable* is general enough (even if the latent variable is not observed). I think that if the scope of the application is properly narrowed down, this study would have enough significance, and the scope should be clearly mentioned in the introduction of the paper.
> >
> > Overall, I believe the theorems and the empirical results would be worth publishing, but their practical applicability should be clearly stated. Therefore, let me keep the original score.

---

### Official Review · Reviewer_TFiH · 2023-07-09

**Soundness:** 3 good
**Presentation:** 3 good
**Contribution:** 3 good
**Rating:** 7
**Confidence:** 3

**Summary:**

This paper studies a unified framework of contrastive learning. The authors extend the theoretical result of [1] to embrace more contrastive losses, adapt it to marginal distributions beyond uniform distribution, and further relax some conditions that are required in [1]. The experimental results support the theoretical claim, and the authors also study/discuss the limitation of the theoretical framework.

[1]: Contrastive Learning Inverts the Data Generating Process

**Strengths:**

Overall, the story telling is good: The authors first come to a theoretical guarantees under some relaxed assumptions, then empirically validate the theoretical findings, and further analyze the impact of the violated assumptions, and discussed the limitations. Compare with previous close works, there are two advantages of this work, include:

1. A unified framework instead of focusing on InfoNCE family, and 2. More relaxed conditions of the theorems.

In order to justify their theorem, the experiments on the synthetic datasets are designed to embrace more contrastive losses, and many scenarios are designed for non convex latent spaces, nonuniform marginals and without conditionally independent assumptions on latents.

The studies on the limitations of the framework also bring some insights for future research.


**Weaknesses:**

This paper has a lot of similarities with [1] (the 56th reference in the draft) in the foundation. Though this work does relax the theoretical guarantee of [1], the authors of [1] also empirically show that the underlying factors of variations can be identified even if the theoretical assumptions are severely violated. In Table 2 of this work, the authors also try to examine their theoretical framework under partially violated assumptions.


**Questions:**

-- Regarding batch size, why the authors use 5120 in line 205, and 4096 in line 268?

-- In this paper, for all the experiments, are users using the original SCL or the modified SCL (line 118-121).

-- It is interesting to the authors conduct ablation studies on the number of latent factors, I have a few questions:

1. What scenario are you using for this experiment?
2. Compare with section 4.1, are you sampling more data? Or, as the dimension increases, are you also exponentially increase the data?

**Limitations:**

No negative impact found

---

> ### Author Rebuttal · Authors · 2023-08-08
>
> We thank the reviewer for the valuable feedback and are glad that the reviewer recognizes the novelty and benefit of our work.
>
> Q1. In Section 4.5 (line 268), we analyzed the impact of larger dimensions on disentanglement performance, which requires more GPU memory. We decided to reduce the batch size to accommodate larger dimensions in memory.
>
> Q2. We used the modified SCL loss throughout the entire paper. Please also see the general response for more details.
>
> Q3.1. The generative process in Section 4.5 is the same as “Box (simple)” in Section 4.1. It differs only in the particular parameter analyzed, for example, the number of dimensions.
>
> Q3.2. In Section 4.1, 4.2 and 4.5 we have basically an infinite amount of data since we sample in each epoch new data from the “unknown” generator. We will make this clearer in the text. For a study of the impact of the number of data in a similar context, we would like to refer to [25].
>
> **References**
>
> [25]: Nonlinear ica of temporally dependent stationary sources

---

> > ### Comment · Area_Chair_W6wf · 2023-08-18
> >
> > Dear Authors,
> >
> > The reviewer did not acknowledge your response, so I am joining the discussion. I have read the review and rebuttal and have no further questions.
> >
> > Kind regards,
> > Your AC

---

> > ### Comment · Reviewer_TFiH · 2023-08-18
> >
> > I don't have more questions for now, thank you for your rebuttal.

---

### Official Review · Reviewer_iM6P · 2023-07-25

**Soundness:** 4 excellent
**Presentation:** 4 excellent
**Contribution:** 4 excellent
**Rating:** 7
**Confidence:** 4

**Summary:**

This paper proves two identifiability theorems for a broad family of contrastive learning methods,  without imposing the commonly used independence constraint. The theoretical findings are validated on several benchmark datasets, and the theoretical and practical limitations are fully discussed.

**Strengths:**

1. The considered contrastive model family is broad enough to include several widely-used contrastive learning losses, in which some are never employed for disentanglement (SCL and NWJ objectives). This provides the paper with adequate novelty.
2. The proofs of the theorems are enlightening for me. In my opinion, the techniques used in the proofs will push this field forward.
3. The theoretical and practical limitations of the theory and the considered models are fully discussed, which is appreciated because the limitations of a work are of great meaning to readers.

**Weaknesses:**

1. If the paper can discuss the physical or geometric picture of the conditions and theory, it will be much better for readers to understand the theoretical results.
2. The are some typos in the paper, e.g. Equation (23) on Page 4 of the Appendix misses a right bracket.
3. Some concepts are used in the paper but not introduced, like 'semi-metric' in Equation (1) and 'norm-induced metric' in Theorem 1. I will suggest the authors briefly introduce these mathematical concepts in the footnotes, which will enhance the readability of the paper.

**Questions:**

1. why $\beta=2$ is excluded in the conditions of Theorem 2?

**Limitations:**

 The authors have adequately addressed the limitations.

---

> ### Author Rebuttal · Authors · 2023-08-08
>
> We appreciate the positive assessment. We will discuss the intuition behind the theory in more detail in the revised version and add footnotes on the used mathematical concepts.
> We also thank the reviewer for pointing out typos.
>
> **Why β=2 is excluded in the conditions of Theorem 2?**
>
> Excluding β=2 is analogous to the assumption of non-Gaussian sources in linear ICA and is also equivalent to the exclusion of quasi-Gaussian sources in [25]. Without further knowledge about the shape of the latent space, the marginals, or the generator, strong identifiability is in this case provably impossible.
>
> **References**
>
> [25] Nonlinear ica of temporally dependent stationary sources

---

> > ### Comment · Area_Chair_W6wf · 2023-08-18
> >
> > Dear Authors,
> >
> > The reviewer did not acknowledge your response, so I am joining the discussion. I have read the review and rebuttal and have no further questions. I suggest explaining this more clearly after the theorem.
> >
> > Kind regards,
> > Your AC

---

> > ### Comment · Reviewer_iM6P · 2023-08-18
> >
> > Thanks for the authors' reply. I have no further questions and I will maintain my score.

---

### Official Review · Reviewer_CnBd · 2023-07-26

**Soundness:** 3 good
**Presentation:** 3 good
**Contribution:** 3 good
**Rating:** 6
**Confidence:** 2

**Summary:**

The paper develops a new theoretical framework for learning disentangled representations with contrastive methods. Building on the theory that contrastive methods approximately invert the data generating process, the paper extend it by proposing a new dissimilarity measure. The proposed framework supports a broader family of contrastive methods and relaxes the assumption about the data distribution and latent independence. Detailed theoretical proof is provided. Experiments validate the effectiveness of the proposed framework.


**Strengths:**

Originality

The paper is an extension of previous work [56] and does provide a new framework which can apply to more CL approaches and requires weaken assumption about both the data distribution and latent independence. I think the differences are obvious and the paper seems novel to me.

Quality

The paper is well supported by theoretical analysis. Codes are also provided.

Clarity

Overall the paper is well-organized.


**Weaknesses:**

The author should provide more intuition and high level explanation of their designs (details in Questions).

**Questions:**

- Could the author provide more intuition about the design of the dissimilarity measure (L106)? Are the optimal $\alpha$ and $\bar{\alpha}$ function unique? Why jointly optimize them with the network help?
- For a more complex dataset in real task, what are the considerations when designing $\alpha$ and $\bar{\alpha}$ . Do you think a simple three-layer neural networks are also enough? Can you explain more on L287? What does "incorrect shapes" mean here?
- It would be better if qualitative visualizations/comparison on more complex dataset (e.g. KITTI Masks or 3DIdent) are added to show the improvements.

---

> ### Author Rebuttal · Authors · 2023-08-08
>
> We thank the reviewer for the positive evaluation. We appreciate the nice summary of our research and the hints on where we can further expand the intuition behind the design choices. Please find below a detailed response to the questions raised.
>
> Q1. Yes, the optimal $\alpha$ and $\tilde{\alpha}$ are unique up to an additive constant and depend only on the distribution of the generating factors (see Eq. 7/8). We add these learnable functions to the similarity measure to ensure that the analytical optimum is reachable. If these functions are not correct, the learned representation could potentially be distorted to compensate for these errors. However, our experiments in Section 4.2 show that even when they do not match the ground truth, the latent factors can still be disentangled.
>
> Q2. More complex marginals (e.g. higher dimensions, stronger correlations) require networks with higher capacity to be learned correctly. A learnable bias (or zero for InfoNCE) corresponds to uniform marginals and is therefore often a reasonable choice in practice. Although this was not observed in our experiments, the addition of more complex $\alpha$ and $\tilde{\alpha}$ can potentially introduce poor local minima, for example if they are initialized far from the ground truth and thus negatively affect the learned representation. This is what we mean with “incorrect shapes”. We speculate that this, the higher memory consumption, and the little gain when compared to assuming uniform marginals are the reasons why not more sophisticated similarity measures have found adoption in practice.
> Our personal advice is to start with a learnable constant and introduce more complex functions (e.g. NNs) in a fine-tuning stage. We would also like to point out that inspecting the learned \alpha and \tilde{\alpha} can yield further insights into the distribution of the unknown latents.
>
> Q3. In our experiments we haven’t observed a clear improvement on disentanglement for the investigated datasets, but also no decreased disentanglement scores. This indicates that the optimal encoder in the constrained function space ($\alpha = \tilde{\alpha} = c$) is very close to the optimal encoder in the unconstrained space. However, to fully understand how the optimal encoder is affected, further research is needed.

---

> > ### Comment · Area_Chair_W6wf · 2023-08-18
> >
> > Dear Authors,
> >
> > The reviewer did not acknowledge your response, so I am joining the discussion. I have read the review and rebuttal and have no further questions.
> >
> > Kind regards,
> > Your AC

---

### Official Review · Reviewer_6CFv · 2023-07-28

**Soundness:** 2 fair
**Presentation:** 3 good
**Contribution:** 2 fair
**Rating:** 4
**Confidence:** 3

**Summary:**

Authors propose a generalized contrastive learning framework extending the guarantees for disentanglement learning. The paper also links contrastive learning objective with identifiability results in both week and strong identifiability setting.

**Strengths:**

- The paper is mostly well written and easy to follow (in few places it is hard to digest)
- Authors do great job in converting all different contrastive learning objectives with unified notation
- contrastive learning was empirically studied for it's effectiveness for disentanglement learning, authors to a great job in formally introducing these concepts

**Weaknesses:**

- Some assumptions may be missing in case of thm1 (ref. questions for details)
- Results on causal3DIndent are hard to interpret (ref. questions)

**Questions:**

Here are some question that makes the paper bit unclear, addressing them can possibly improve the quality of the paper:

Q1.  To recover true latent (to establish identifiability) you need contrastive examples varying in only one particular generating factor, which might not be always possible in reality, are there any tricks to circumvent this issue

Q2. In case of theorem 1, I believe there are some strict constraints on how \tilde{s} is constructed using s, these assumptions are not explicitly mentioned. (if there are no constraints, it's unclear how do you account for confounded generating factors)

Q3. In case of confounded datasets, how would your identifiability result still hold? Since, in proof sketch you do not have any assumption over sufficiency and confounding factors.

Q4. homeomorphism condition implicitly assumes the function to be bijective, continuous, and invertible which I believe is stronger assumption that [27] (correct me if I misunderstood the contribution).

Q5.  The experiments are not clearly detailed: how is the data generated? what are the data generating factors?

Q6. why is R2 value >> MCC? : as higher R2 indicates higher correlation which should result in higher MCC, but opposite effect is observed in your experiments, can you provide justification for this behavior?

Q7. More details on training procedure would be helpful (how is parameter b learned?)

Q8. How does the bound on features affect correlation values? (based on table 4 results bounded MCC >> unbounded MCC, what could be possible reason for this? is it an artifact of clipping values?)


**Limitations:**

Few technical limitations are mentioned in the paper, limitations of contrastive data acquisition in real-world setting is not discussed.

---

> ### Author Rebuttal · Authors · 2023-08-08
>
> We thank the reviewer for the detailed feedback and advice on how to improve the quality of the work. We will include the mentioned limitation in the revised version and are happy to provide clarification on the raised issues.
>
> The reviewer assessed the present work from the perspective of causal representation learning, however, there seems to be a misunderstanding regarding the modeling of the generative process. We do not explicitly model causal relationships between the latent factors in this paper. Instead, we assume that the latent factors of positive pairs are related according to Eq. 1. This allows for correlations between latent factors (no assumption on $p(s)$ except the support must be connected), and they need not be conditionally independent because $Q(\tilde{s})$ does not necessarily factorize. Nevertheless, the connection to causal representation learning is an interesting research direction.
>
> Q1. In our approach, it is not necessary that only one generating factor changes between instances in a positive pair. In Theorem 1, the generating factors may also not be one-dimensional, such as rotations in 3D.
>
> Q2. In Theorem 1 we assume that \tilde{s} depends on s according to Eq. 1. Roughly speaking, by considering pairs, the correlations between different latent factors cancel each other out.
>
> Q3. It is not clear to us how confounders would invalidate our theoretical results. If our previous answers cannot clarify this question, perhaps the reviewer is willing to elaborate on the question or provide an illustrative counterexample?
>
> Q4. The learned encoder in [27] is assumed to be a smooth homeomorphism with smooth inverse, i.e., a diffeomorphism (condition 5 in Theorem 1). We lift these assumptions and only require the encoder to be differentiable.
> Under the stated assumptions in Theorem 1, we can show that the optimal encoder must also be invertible. It is thus not necessary to restrict the function space.
>
> Q5. The latent factors for 3DIdent are position, rotation, spotlight position and color of the object, spotlight and background  (3+3+1+3=10). For KITTI Masks they are the object position in the image and scale (2+1=3). In the other synthetic experiments the latent factors don’t have a particular meaning. Please also see the general response.
>
> Q6. We use R2 to measure weak identifiability (up to affine transformations) and MCC for strong identifiability (up to permutations and element-wise transformations). If, for example, the mapping from the true to the learned latents is a rotation by 45° around some axis, the R2 value will be 1 since there exists a linear map between them, but the MCC score will be much lower because the axes are not aligned.
>
> Q7. The scaling parameter $b$ is only learned when the output space $Z$ is restricted to a scalable box, i.e., $Z=[0,b]^n$. We achieve this by multiplying the output of a sigmoid in the last layer with b. In this case b is treated like any other learnable parameter in the neural network.
>
> Q8. When the true latent space is unbounded and the conditional of the generating factors is (quasi-)Gaussian or the chosen distance metric in the contrastive loss is Euclidean, we cannot achieve strong identifiability because the loss is then rotationally invariant. In this case we can identify the generating factors only up to affine transformations (high R2 scores). However, if the true latent space is a box, the conditional distribution is truncated and thus not rotationally invariant anymore. When we also restrict the learned latent space to a box these boxes align under certain conditions, leading to higher MCC scores. This is only possible through additional knowledge about the latent space.

---

> > ### Comment · Reviewer_6CFv · 2023-08-14
> >
> > Thank you for the thoughtful rebuttal; I have no further questions

---

### Author Rebuttal · Authors · 2023-08-08

We thank all reviewers for their constructive feedback and overall positive evaluation. Below we describe the main concerns and how we intend to address them. Additional points are discussed in the individual responses to the reviewers.

**Confusion about the generative process**

We will add a clearer description of the generative process in the revised version. We generate positive pairs by first sampling $s$ from a marginal distribution $p(s)$ and then $\tilde{s}$ from a conditional distribution according to Eq. 1. The observations are then obtained by transforming the true latents with an invertible generator. The latent distribution and generator are typically unknown in practice.

**More intuition behind Theorems 1 & 2**

Our main assumption (Eq. 1) essentially states that the probability of observing a positive pair decreases exponentially with the distance between the corresponding latents, "contaminated" only by $Q(\tilde{s})$ and the marginal $p(s)$. Neither $Q(\tilde{s})$ nor $p(s)$ depends on both instances of the pair. This allows us to learn a representation that preserves this distance metric together with $p(s)$ and $Q(\tilde{s})$. In this way we can identify the latents up to isometries. By further restricting the dimension and connectivity of the space, such an isometry must be affine and even a generalized permutation matrix for certain metrics.

**Transferability of theory to practice**

There are several ways to create positive pairs for contrastive learning: data augmentation, adjacent points in sequences, multiple views or modalities, weak supervision, and more. Each of these mechanisms can be described in terms of the underlying latent distribution. Our theoretical findings provide identifiability guarantees of the generative factors for specific forms of this mechanism (Eq. 1). Although this equation applies only to some practical scenarios, we believe it contributes to our general understanding. The study of other forms is an interesting future research direction.

**Comparison between original and modified SCL**

Throughout the paper, we use our modified version of SCL instead of the original version. This modified version sometimes fails to identify the latent factors due to numerical instabilities that the original SCL does not have (e.g., less oscillation during training). These instabilities are indeed due to the additional exponential term. However, in further experiments (which we will include in the appendix), the original SCL performs worse in the selected scenarios based on R2 and MCC. This does not mean, however, that these learned representations are generally less useful, only that the relationship between the true and inferred latents is not affine.

---

### Decision · Program_Chairs · 2023-09-21

**Decision:**

Accept (poster)

**Comment:**

The paper is well-written and easy to follow, with the main contribution (the unified framework) being accessible and relevant. The reviewers positively called out the proof technique, which they found interesting and likely to spark more contributions. Some concerns were raised about the gaps in assumptions between theory and practice. I would recommend the authors to more clearly point this out in the paper.